# A Stable Large-Scale Multiobjective Optimization Algorithm with Two Alternative Optimization Methods

**DOI:** 10.3390/e25040561

**Published:** 2023-03-25

**Authors:** Tianyu Liu, Junjie Zhu, Lei Cao

**Affiliations:** College of Information Engineering, Shanghai Maritime University, Shanghai 201306, China

**Keywords:** evolutionary algorithms, large-scale multiobjective optimization, two alternative optimization methods, Bayesian-based parameter adjusting

## Abstract

For large-scale multiobjective evolutionary algorithms based on the grouping of decision variables, the challenge is to design a stable grouping strategy to balance convergence and population diversity. This paper proposes a large-scale multiobjective optimization algorithm with two alternative optimization methods (LSMOEA-TM). In LSMOEA-TM, two alternative optimization methods, which adopt two grouping strategies to divide decision variables, are introduced to efficiently solve large-scale multiobjective optimization problems. Furthermore, this paper introduces a Bayesian-based parameter-adjusting strategy to reduce computational costs by optimizing the parameters in the proposed two alternative optimization methods. The proposed LSMOEA-TM and four efficient large-scale multiobjective evolutionary algorithms have been tested on a set of benchmark large-scale multiobjective problems, and the statistical results demonstrate the effectiveness of the proposed algorithm.

## 1. Introduction

Nowadays, optimization problems with large-scale decision variables appear in various fields, such as artificial intelligence [1], big data mining [2], large-scale software engineering [3], economic decision-making problems [4,5], and so on. Large-scale multiobjective optimization problems (LSMOPs) are generally referred as multiobjective optimization problems involving hundreds or even thousands of decision variables [6,7]. However, studies have shown that the traditional multiobjective evolutionary algorithms, such as MOEA/D [8], NSGA-III [9], and MOPSO [10], tend to converge slowly when solving LSMOPs. This phenomenon is mainly because the volume of decision variable space grows exponentially when the number of decision variables increases. In this case, searching for Pareto optimal solutions becomes very difficult. This phenomenon is called the curse of dimensionality [11,12].

Recently, researchers have proved that the idea of “divide and conquer”, which has been widely adopted in cooperative coevolution (CC), can efficiently address LSMOPs [13]. The core idea of “divide and conquer” is to divide the large-scale decision variables in LSMOPs into multiple lower-dimensional groups, thereby transforming LSMOPs into multiple small-scale problems. After that, conventional evolutionary algorithms (EAs), such as the genetic algorithm (GA), particle swarm optimization (PSO), and differential evolution algorithm (DE), can be used as efficient tools to solve the transformed small-scale problems. Please note that conventional EAs cannot be used to solve LSMOPs directly because of the “curse of dimensionality”. The main difference between large-scale multiobjective optimization algorithms (LSMOEAs) and conventional EAs is that various strategies have been incorporated into LSMOEAs to solve the “curse of dimensionality” caused by large-scale decision variables.

Among various large-scale multiobjective optimization algorithms, LSMOEAs based on decision-variable grouping strategies have received more and more attention. Most LSMOEAs based on decision-variable grouping strategies divide the decision variables into convergence-related and diversity-related variables [12]. Specifically, the convergence-related variables help LSMOEAs find the solution sets closer to the ideal solution sets. In contrast, the diversity-related variables help LSMOEAs find the solution sets with a better distribution. Existing decision-variable grouping strategies can be divided into fixed grouping strategies [13,14,15,16,17,18,19,20] and dynamic grouping strategies [21,22,23,24]. In a fixed grouping strategy, the grouping results do not change during the evolution process, i.e., the evolutionary algorithm for large-scale many-objective optimization, LMEA. In the dynamic grouping strategies, the grouping results change during the optimization process [12], i.e., cooperative coevolutionary generalized differential evolution 3, CCGDE3. The quality of the grouping of decision variables affects the performance of algorithms directly [24]. Therefore, grouping decision variables reasonably and effectively is a challenging problem for LSMOEAs based on grouping strategies.

For LSMOEAs with fixed grouping strategies, the grouping results of decision variables are obtained according to the initial population, whose members are the initial solutions for the optimized problem and are usually generated randomly in the decision-variable space. Since the grouping results will not change in LSMOEAs with fixed grouping strategies, the corresponding algorithm may obtain unsatisfactory optimization performance if the grouping results are not good enough [12]. Figure 1 gives the grouping results of LMEA, a representative LSMOEA with a fixed grouping strategy, on the benchmark test problem BT6 [25]. In LMEA, the division of decision variables is implemented on the initial population, and the group results are used in the optimization process. It can be observed from Figure 1 most decision variables (29 out of 30) are regarded as convergence-related variables, and only one decision variable is the diversity-related decision variable. Therefore, LMEA will likely obtain a Pareto optimal set that cannot distribute well on the Pareto front of BT6. In Figure 2, the black curve represents the ideal Pareto front, which demonstrates the position of the ideal solutions for BT6. The gray dots are the members (solutions) of the current population. The gray dots should be distributed as evenly as possible on the Pareto front to guarantee population diversity. However, in LMEA, the population focuses on a single point on the Pareto front as the evolutionary process continues (as shown in Figure 2c), which demonstrates that the LMEA fails to maintain population diversity at the last stage of the evolution process. This is because the grouping results based on the initial population are sometimes unreasonable, i.e., there are too many convergence-related decision variables in BT6 (as shown in Figure 1). In this case, using the fixed group results based on the initial population to guide the evolution process cannot achieve the balance between convergence and population diversity.

This paper proposes a large-scale multiobjective optimization algorithm with two alternative optimization methods (LSMOEA-TM) to deal with LSMOPs, and the main new contributions are summarized as follows.
(1)In the proposed two alternative optimization methods, two group strategies, namely, the convergence-related grouping strategy and the diversity-related grouping strategy, are introduced to group the large-scale decision variables based on the evaluation of the population. Specifically, if there is a significant performance degradation in the current population, the diversity-oriented stage is implemented by adopting the diversity-related grouping strategy. Suppose the diversity-oriented stage has been carried out for a certain number of generations. In that case, LSMOEA-TM implements the convergence-oriented stage with the help of the convergence-related grouping strategy.(2)A Bayesian-based parameter adjustment strategy is proposed to modify the parameters in the convergence-related and diversity-related grouping strategies to reduce the computational cost of the proposed algorithm.

The rest of this article is organized as follows. In Section 2, the representative algorithms for solving LSMOPs are given. Section 3 introduces the proposed LSMOEA-LS in detail. Section 4 shows experimental results and the analysis of LSMOEA-TM and four efficient LSMOEAs. Section 5 gives the concluding remarks.

## 2. Background

### 2.1. Large-Scale Multiobjective Optimization Problems (LSMOPs)

The problems having multiple conflicting objectives are called multiobjective optimization problems (MOPs) [8]. An MOP can be formulated as shown in Equation (1), where x= ($x_{1},x_{2},\ldots ,x_{D}$) is a candidate solution, M is the number of objectives, and D is the number of decision variables. If D≥100, then the MOP is regarded as a large-scale MOP, namely, an LSMOP.
(1)$$\left\{\begin{array}{c} min\;F\left(x\right)=\left\{f_{1}\left(x\right),f_{2}\left(x\right),\ldots ,f_{M}\left(x\right)\right\}\\ subject\;to:x\in \Omega^{D} \end{array}\right.$$

Finding one solution that can optimize all objectives simultaneously is impossible since the objectives are mutually conflicting for MOPs [26]. Suppose a minimized MOP, the dominant relation between solutions x and y can be obtained according to Equation (2): (2)$$x≺y\;if\;f\;\forall f_{i}\left(x\right)\leq f_{i}\left(y\right)\;\left(i=1,2,\ldots ,M\right)\wedge \;\exists f_{j}\left(x\right)<f_{j}\left(y\right)$$

If solution $x^{*}$ cannot be dominated by any other solutions, then $x^{*}$ is the nondominated solution. For MOPs, the optimization algorithms aim to find a set of nondominated solutions (Pareto optimal solutions). Recently, multiobjective evolutionary algorithms (MOEAs) have become efficient tools for handling MOPs [8,10]. However, the performance of MOEAs may degrade when extended to solve LSMOPs [12]. This is because the decision variable space increases dramatically with the increase of the number of decision variables. Therefore, it is difficult for traditional MOEAs to find promising nondominated solutions when handling LSMOPs.

### 2.2. Large-Scale Multiobjective Evolutionary Algorithms (LSMOEAs)

Most existing LSMOEAs are based on “divide and conquer”. The core idea of “divide and conquer” is to divide a complex problem into several smaller subproblems and then obtain the solution of the original problem by collating the solutions of the subproblems. Based on the idea of “divide and conquer”, an LSMOP can be divided into multiple small-scale multiobjective optimization problems. The original LSMOP can be obtained by integrating the solution of the transformed small-scale multiobjective optimization problems. The existing LSMOEAs can be mainly divided into the following categories.

#### 2.2.1. LSMOEAs Based on Fixed Grouping

The main idea of the fixed grouping strategy is to adopt fixed grouping results of the decision variables in the evolution process. Ma et al. [14] proposed a multiobjective evolutionary optimization algorithm based on a decision-variable analysis, namely, MOEA/DVA, to solve LSMOPs. In MOEA/DVA, the decision variables of some individuals were perturbed, and the individuals were randomly selected from the initial population. Then, the decision variables were divided into convergence-related variables and diversity-related variables according to the dominance relationship among the perturbed individuals. Based on MOEA/DVA, a new fixed grouping strategy, called LMEA [16], was proposed. This strategy divided the decission variables according to the angles between the perturbation fitting lines and the hyperplane normal lines. Cao et al. [20] proposed a new large-scale multiobjective optimization algorithm called mogDG-shift and adopted a graph-based differential grouping strategy to decompose the large-scale decision variables. 

The fixed grouping strategy can achieve a stable grouping result since it adopts fixed grouping results of the decision variables in the evolution process. However, most fixed grouping strategies get grouping results based on the initial population, and the initial population sometimes cannot reflect the characteristics of the decision-variable space. Therefore, the fixed grouping strategy, implemented on the initial population, may generate bad group results and lead to unsatisfactory optimization performance.

#### 2.2.2. LSMOEAs Based on Dynamic Grouping

Antonio and Coello proposed a classic LSMOEA based on dynamic grouping, namely, CCGDE3 [21]. The main idea of CCGDE3 was to randomly divide the decision variables into several equal groups. Subsequently, Antonio and Coello replaced the third generation generalized differential evolution (GDE3) in CCGDE3 with MOEA/D [8] and proposed a new algorithm MOEA/D2 [22]. However, the grouping size was set to a fixed value in CCGDE3 and MOEA/D2. Therefore, these two algorithms may not adapt well to different types of LSMOPs. To deal with the problem mentioned above, MOEA/D-RDG (MOEA/D combining random-based dynamic grouping) was proposed. In MOEA/D-RDG, a grouping parameter pool was constructed to improve the generalization ability of the proposed algorithm. 

Nowadays, dynamic grouping strategies are applied to solve various LSMOPs successfully. However, many of the existing dynamic grouping strategies are derived from random grouping, which may lead to unstable and unsatisfactory group results, especially for multiobjective problems with more than 1000 decision variables and multiobjective problems with complex relationships between decision variables [12].

For LSMOEAs with dynamic grouping strategies, the grouping results of decision variables can adapt to the evolving population. However, dynamic grouping strategies cannot guarantee the stability of grouping results, especially for random-based dynamic grouping strategies. Figure 3 demonstrates the IGD values obtained by CCGDE3 and LMEA over 30 independent runs on two benchmark problems, namely DTLZ2 and UF6. CCGDE3 and LMEA are the typical algorithms in LSMOEAs with dynamic and fixed grouping strategies, respectively. IGD is a widely used metric that can evaluate a multiobjective algorithm’s performance from the perspective of both convergence and diversity. As Figure 3 shows, the performance of CCGDE3 is more unstable than that of LMEA.

#### 2.2.3. Other LSMOEAs

Besides the LSMOEAs based on fixed and dynamic grouping strategies, many other large-scale multiobjective optimization algorithms exist. A typical representative algorithm is a large-scale multiobjective algorithm based on problem reconstruction (LSMOF) [27]. The core idea of LSMOF is to transform the original LSMOP into a small-scale single-objective optimization problem through problem reconstruction. In DGEA [28] and S3-CMA-ES [29], the population was divided into multiple subpopulations, and each sub-population focused on a single-objective function. Chen et al. [30] proposed an inverse-modeling multiobjective evolutionary algorithm based on decomposition (IM-MOEA/D) to solve LSMOPs. IM-MOEA/D divided the decision variable space into several subareas by reference vectors. Then, the population was divided into several groups according to their distribution in the decision-variable space, and each group evolved separately. The solutions of all groups cooperated in obtaining promising results for LSMOPs. In [31], a fuzzy decision variable framework with various internal optimizers (FDV) was proposed. In FDV, a two-stage optimization strategy containing the rough optimization stage and the fine optimization stage was introduced to handle LSMOPs efficiently. 

It can be found that the algorithms mentioned above also adopt the idea of “divide and conquer”. However, in these algorithms, the division process focuses on objective spaces or populations rather than decision variables.

## 3. Method

Figure 4 shows the main framework of LSMOEA-TM. LSMOEA-TM adopts two alternative optimization methods, which include two optimization stages: the convergence-oriented stage and the diversity-oriented stage. As shown in Figure 5, the two stages adopt different grouping strategies to guide the evolution process. As shown in Figure 4, LSMOEA-TM chooses the appropriate optimization stage adaptively to achieve the balance of convergence and population diversity. Specifically, if there is a significant performance degradation in the current population, the diversity-oriented stage is implemented by adopting the diversity-related grouping strategy. In Figure 4, newHV and oldHV are the values of the hypervolume (HV) metric [32] of the current and the last population, respectively. Suppose the diversity-oriented stage has been carried out for a certain number of generations. In that case, LSMOEA-TM implements the convergence-oriented stage with the help of the convergence-related grouping strategy. In LSMOEA-TM, the convergence-oriented and diversity-oriented stages are executed alternatively to balance convergence and population diversity. 

Furthermore, to reduce computational costs, this paper introduces a Bayesian-based parameter adjusting strategy to modify the parameters in the convergence-related and diversity-related grouping strategies. As shown in Figure 5, the two stages have a similar structure. The difference between the two stages is that they use a convergence-related strategy and a diversity-related grouping strategy, respectively.

It can be observed from Figure 4 that the main operators in LSMOEA-TM are the initialization of the population and repository population, the Bayesian-based parameter adjustment, and the two alternative optimization methods. Detailed descriptions of the key operators are given below.

### 3.1. Initialization

As shown in Figure 6, the population (POP) can be initialized as an N×D matrix and N is the population size. In POP, each row is a candidate solution that has D decision variables, as shown in Figure 6. Each element of one candidate solution is generated from the search range randomly. The repository population (REP) contains the nondominated solutions found by the algorithm. In the initialization step, REP contains the nondominated solutions in POP. 

### 3.2. Two Alternative Optimization Methods

In LSMOEA-TM, the two alternative optimization methods contain two stages: the convergence-oriented stage and the diversity-oriented stage. As shown in Figure 4, the difference between the two stages is that they adopt different grouping strategies to divide decision variables. In LSMOEA-TM, at first, the convergence-oriented stage is adopted to update POP. Then, LSMOEA-TM chooses the appropriate optimization stage according to the evaluation of POP. In Figure 4, newHV and oldHV are the values of the hypervolume (HV) metric of the current and the last POP, respectively. The HV is a commonly used metric to assess the performance of a multiobjective algorithm. The reason for choosing the HV metric here is that the ideal nondominated solutions to problems are not needed when calculating the HV, which is given in [32]. For the HV, the larger the value, the better the quality of the solutions. $\left(newHV-oldHV\right)/oldHV>\varepsilon$ means the updated POP is better than the last POP. In this case, LSMOEA-TM continues to perform the convergence-oriented stage by adopting the convergence-related grouping strategy. Otherwise, LSMOEA-TM performs a diversity-oriented stage by adopting the diversity-related grouping strategy. To balance the population diversity and convergence, LSMOEA-TM switches to implementing the convergence-oriented stage after performing the diversity-oriented stage for s times. A specific description of the update of POP and REP, the Bayesian-based parameter adjustment, the convergence-related, and the diversity-related grouping strategy is presented below.

#### 3.2.1. Update of POP and REP

The update of POP is designed based on the grouping results of decision variables. Algorithm 1 gives the detailed procedure for the update of POP. As shown in Section 3.2.1 and Section 3.2.2, the decision variables are divided into convergence-related variables (CV) and diversity-related variables (DV). In Algorithm 1, the update of POP contains the convergence-related update stage and the diversity-related update stage, which are implemented on the convergence-related and diversity-related variables, respectively. In the convergence-related update stage, the parent individuals are selected according to the nondominated ranks of individuals in POP by the tournament selection method (line 1 in Algorithm 1). After that, the crossover and mutation operators are carried out only on the convergence-related variables of the parent individuals to obtain the offspring individuals (line 2 in Algorithm 1). If the offspring individuals are better than (dominate) the parent individuals, then replace the parent individuals with the offspring individuals (lines 3–4 in Algorithm 1). Based on the POP1 obtained in the convergence-related update stage, the diversity-related update stage is implemented only on the diversity-related decision variables (lines 5–21 in Algorithm 1). It can be observed from Algorithm 1 that there are two main differences between the convergence-related and diversity-related update stages. The first one lies in the selection of the parent population. In the diversity-related update stage, the parent population is selected from the population randomly (line 5 in Algorithm 1). The second difference is that the diversity-related update stage obtains the updated population (new_POP) by selecting N individuals from Candidates with consideration of both nondominated ranks and angle distances (line 8–21 in Algorithm 1). As shown in Algorithm 1, N is the size of POP. The angle distance aims to help the algorithm obtain the updated population that distributes dispersedly.

Since the REP stores the nondominated solutions found by the algorithm so far, the update of REP can be achieved by selecting the nondominated members from the update POP and the original REP.
**Algorithm 1:** Update of POP**Input:** population POP; convergence-related decision variables CV**;**   diversity-related decision variables DV**Output:** updated population new_***POP***1.Select the parent population from ***POP*** by the tournament selection method according to the nondominated ranks of the individuals in ***POP***;2.Obtain offspring population ***Q*** by performing the crossover and mutation operators only on the ***CV*** of the parent population;3.Candidates←POP∪ ***Q***, obtain the nondominated ranks of the members in Candidates.4.Select N individuals with the better nondominated ranks from Candidates as *POP*1 and *N* is the size of *POP*;5.Select the parent population from ***POP*1** randomly;6.Obtain offspring population ***Q*** by performing the crossover and mutation operators only on the ***DV*** of the parent population;7.Candidates←POP∪ ***Q***, obtain the nondominated ranks of the members in Candidates.8.new_*POP = *$\left\{x|x\in Candidates\;\mathrm{and}\;nondominate\_rank\left(x\right)=1\right\}$. If |new_*POP| > N*, then delete members from new_*POP* randomly until |new_*POP| = N*.9.*R* = 2; 10.While |new_*POP| < N*11.  P contains the members from Candidates whose nondominated ranks is *R*;12.  If |new_*POP*|+|*P*| > *N*13.While |new_*POP| < N*14.        new_*POP*$\leftarrow new\_POP\cup {\mathrm{argmax}}_{x\in P}\left(\underset{y\in new\_POP}{\min}\mathrm{angle}\left(x,y\right)\right)$;15.        Delete *x* from *P;*16.EndWhile17.  Else18.        new_
*POP*
←new_POP∪
*P*19.  EndIf20.  *R* = *R* + 1;21.EndWhile

#### 3.2.2. Bayesian-Based Parameter Adjustment Strategy

As described in Section 3.2.1 and Section 3.2.2, both the convergence-related and diversity-related grouping strategies need two parameters, namely nSel and nPer. nSel determines the number of individuals that are selected from POP to be perturbed. nPer determines the number of perturbations for a decision variable to be grouped. Therefore, there are nSel×nPer perturbations to group one variable. If nSel and nPer are too large, it will take too many computations. However, if nSel and nPer are too small, the grouping results may be inaccurate. In this paper, a Bayesian-based parameter adjustment strategy is introduced to obtain appropriate values for nSel and nPer to achieve a balance between grouping accuracy and computations. More specifically, with the help of the Bayesian-based parameter adjustment strategy, the convergence-oriented and diversity-oriented stages can get the group results of decision variables with less computational cost, as shown in Figure 5. The main steps of the Bayesian-based parameter adjustment strategy are as follows:

Step 1: Determine the form of the loss function as shown in Equation (3);

Step 2: Generate a certain number of the initial observation values for nSel and nPer and obtain the loss function values for the generated observation values;

Step 3: Estimate the value of the loss functions for observation samples by the probability surrogate model in Equation (4).

Step 4: Obtain the next values for nSel and nPer by the acquisition function in Equation (5).

In Step 1, the loss function is formulated as shown in Equation (3). The proposed parameter adjustment strategy aims to obtain the value of nSel and nPer as small as possible with acceptable grouping results. In Equation (3), CV and DV represent the number of convergence-related variables and diversity-related variables obtained by the grouping strategies with parameters nSel and nPer, respectively. If CV is much larger than *DV*, then the algorithm may obtain a Pareto-optimal set that concentrates on a small area of the ideal Pareto fronts. If DV is much larger than CV, then the algorithm may have poor convergence. To balance the convergence and diversity, CV and DV should be as close as possible [12]. In Equation (3), $\frac{CV\left(nSel,nPer\right)+DV\left(nSel,nPer\right)}{CV\left(nSel,nPer\right)DV\left(nSel,nPer\right)}$ is used to evaluate the rationality of the grouping results quantitatively. $\theta \left(nSel,nPer\right)$ is the regularization term of parameters nSel and nPer is used to make the grouping parameters as small as possible when decision variables are reasonably grouped. As shown in Equation (3), the L2 parameter regularization method is adopted to obtain $\theta \left(nSel,nPer\right)$. λ is set to $\frac{2}{\left(nSel_{\max}^{2}-nSel_{\min}^{2}\right)+\left(nPer_{\max}^{2}-nPer_{\min}^{2}\right)}$ in order to make θ vary between 0 and 1.
(3)$$\left\{\begin{array}{c} f\left(nSel,nPer\right)=\frac{CV\left(nSel,nPer\right)+DV\left(nSel,nPer\right)}{CV\left(nSel,nPer\right)DV\left(nSel,nPer\right)}+\theta \left(nSel,nPer\right)\\ \theta \left(nSel,nPer\right)=\frac{\lambda }{2}\left[\left(nSel^{2}-nSel_{\min}^{2}\right)+\left(nPer^{2}-nPer_{\min}^{2}\right)\right] \end{array}\right.$$

In Step 2, a certain number of initial observation values of nSel and nPer are generated. For a better illustration, nSel is taken as an example since the generation process for the observation values of nSel and nPer are similar. Suppose ${\mathrm{nSel}}_{\min}$ and ${\mathrm{nSel}}_{\max}$ are the lower and upper limit values of the parameter nSel, then the initial values of nSel can be obtained by a sampling method based on dichotomy. Specifically, the original interval $\left[{\mathrm{nSel}}_{\min},{\mathrm{nSel}}_{\max}\right]$ is first divided into two intervals $\left[{\mathrm{nSel}}_{\mathrm{lower}},\frac{{\mathrm{nSel}}_{\mathrm{lower}}+{\mathrm{nSel}}_{\mathrm{upper}}}{2}\right]$ and $\left[\frac{{\mathrm{nSel}}_{\mathrm{lower}}+{\mathrm{nSel}}_{\mathrm{upper}}}{2},{\mathrm{nSel}}_{\mathrm{upper}}\right]$. Then, two observation values of nSel are generated from the two intervals randomly. The steps mentioned above are repeated until there are enough observation values for nSel. 

In Step 3, a probability surrogate model $f^{*}\left(nSel,nPer\right)$, as shown in Equation (4), is adopted to estimate the value of the loss function $f\left(nSel,nPer\right)$ since calculating the loss function cost more computations than the probability surrogate model. As shown in Equation (4), the Gaussian mixture model, a mixture of K Gaussian distributions, was adopted in this section to estimate the loss function. The Gaussian mixture model was chosen because it can simulate any function distributions theoretically [33]. In Equation (4), $\sum_{i=1}^{K}\alpha_{i}=1$ and $\phi \left(nSel,nPer|\theta_{i}\right)\sim N\left(\mu_{i},\sigma_{i}^{2}\right)$.
(4)$$f^{*}\left(nSel,nPer\right)=\overset{K}{\underset{i=1}{\sum }}\alpha_{i}\phi \left(nSel,nPer|\theta_{i}\right)$$

In Step 4, the acquisition function is implemented to obtain the appropriate values for nSel and nPer. Since the expectation improvement is usually adopted as the acquisition function in Bayesian optimization [34], the expectation improvement, shown in Equation (5), was used to generate the new values for nSel and nPer. The generated values of nSel and nPer are utilized in the grouping strategies to get the grouping results of decision variables. Furthermore, the new generated nSel and nPer and their loss function values are adopted to obtain the probability surrogate model in the next iteration of the Bayesian-based parameter adjustment strategy. Therefore, as the evolution progresses, more observation values are generated, and the probability surrogate model becomes closer to the real loss function.
(5)$$\left\{\begin{array}{c} EI_{f^{*}}=\int_{-\infty }^{+\infty }max\left(\eta -f^{*},0\right)p\left(f^{*}|nSel,nPer\right)d\left(f^{*}\right)\\ \left(nSel_{new},nPer_{new}\right)={\mathrm{argmax}}_{\left(nSel,nPer\right)}EI_{f^{*}} \end{array}\right.$$

#### 3.2.3. Convergence-Related and Diversity-Related Grouping Strategy

As discussed above, the grouping results of the decision variables directly influence the performance of LSMOEAs based on grouping strategies. This paper uses two different grouping strategies, carried out alternatively during the optimization process, to get efficient group results for LSMOEA-TM.

In a convergence-related strategy, decision variables are grouped based on the perturbation results of the selected individuals from POP. Suppose nSel individuals are randomly selected from the population and each decision variable is perturbed nPer times by sampling in the decision-variable space. For a two-objective problem, Figure 7 demonstrates the perturbation results of three decision variables, i.e., $x_{1}$, $x_{2}$, and *x*_3_. In Figure 7, the dotted line *L* is the normal line of a hyperplane $\sum_{i=1}^{m}f_{i}=1$ and m=2. In that case, L indicates the convergence direction for a minimized two-objective problem. If nSel=2 and nPer=10, then 2 individuals are selected to be perturbed and each is perturbed 10 times. For example, the value of $x_{1}$ in the 2 selected individuals is perturbed 10 times to get the perturbation results for $x_{1}$, which can be fitted by a straight line, as shown in Figure 7. After that, the angle of each fitting line and L can be calculated. Therefore, the smaller the angle, the more likely the variable is related to convergence. In Figure 7, each variable can be positioned in an angle space with two dimensions since nSel=2. With the help of the K-means clustering method, all variables can be grouped into two clusters. The cluster with smaller angles contains convergence-related variables, and the other cluster contains diversity-related variables. In Figure 7, $x_{1}$ has relatively large angle values in the angle space. Therefore, it is divided into diversity-related variable group. Conversely, $x_{3}$ is regarded as a convergence-related variable with small angle values. $x_{2}$ is divided into the convergence-related variable group, since it is closer to $x_{3}$ in the angle space. However, as shown in Figure 7, the trajectory of the perturbations of $x_{3}$ is nonlinear. In this case, using a straight line to fit the perturbation results of $x_{3}$ is inappropriate. It can be observed in Figure 7 that $x_{3}$ is more likely to contribute to helping maintain population diversity. Therefore, for some variables with nonlinear perturbation curves, the convergence-related grouping strategy may obtain wrong classification results and lead to unstable optimization performance. To compensate, the following diversity-related grouping strategy is adopted in this paper.

To overcome the shortcomings in convergence-related strategy, the diversity-related grouping strategy is based on the dominance analysis of the perturbation results of variables. As shown in Figure 7, the perturbation results of $x_{3}$ are mutually dominated. Therefore, $x_{3}$ is regarded as a convergence-related variable. For $x_{1}$, the perturbation results are nondominated solutions. Thus, $x_{1}$ contributes more to maintaining population diversity and is regarded as a diversity-related variable. For $x_{2}$, there are both dominated and nondominated relationships in its perturbation results. Therefore, $x_{2}$ contributes to both diversity and convergence. In the diversity-related grouping strategy, the decision variables contributing to diversity and convergence are classified as diversity-related variables. In this case, the algorithm using the diversity-related grouping strategy emphasizes maintaining population diversity in its evolution process. 

## 4. Results

### 4.1. Test Suites and Algorithms to Be Compared

In this section, five test suites consisting of DTLZ [35], WFG [36], UF [37], BT [25], and LSMOP [7] were chosen to evaluate the performance of the proposed LSMOEA-TM. DTLZ, WFG, and UF are the benchmark test suites that are widely used to evaluate the performance of algorithms for solving LSMOPs. BT contains a set of biased test problems, which complicates maintaining population diversity for the evaluated algorithms. For DTLZ, WFG, UF, and BT, the decision variables were set to 100, 500, and 1000, respectively. LSMOP [7] is a test suite proposed recently and contains nine large-scale multiobjective problems (LSMOP1~LSMOP9). For all problems in LSMOP, the number of decision variables was set to 100 × M, where M is the size of the objective space. LSMOP1~LSMOP9 had 300 decision variables since they were all triobjective problems.

To verify the performance of the proposed algorithm, LSMOEA-TM was compared with four efficient large-scale multiobjective optimization algorithms:(1)MOEA/D2 [21], which is a representative algorithm based on the dynamic grouping strategy.(2)LMEA [16], which is a representative algorithm based on the fixed grouping strategy.(3)IM-MOEA/D [30], which uses a decomposition-based strategy to solve LSMOPs.(4)FDV [31], which utilizes a fuzzy search strategy to group decision variables when solving LSMOPs.

### 4.2. Experiment Setting and Measurement Methodology

All experiments in this paper were implemented in MATLAB R2020b on a desktop with a 3.60GHz Intel I Core I i9-9900kf CPU and Windows 11 64-bit operating system with 32 GB of RAM.

For a fair comparison, MOEA/D2, LMEA, IM-MOEA/D, and FDV adopted the recommended parameter settings from [21], [16], [30], and [31], respectively. For LSMOEA-TM proposed in this paper, the strategy switch threshold ε (Section 3.2) was set to −0.15, and the threshold value s (Section 3) was set to 3 empirically. All five algorithms adopted the simulated binary crossover (SBX) operator and the polynomial mutation operator to generate offspring. The crossover probability was $p_{c}=1.0$, and the mutation probability was $p_{m}=1/D$, where D is the number of decision variables. For all algorithms, the population size was 100, and the maximum number of function evaluations was set to 1,000,000, 5,000,000, 15,000,000, 30,000,000, and 50,000,000 for the test problems with 100, 500, 1000, 2000, and 4000 decision variables, respectively.

In this paper, two widely used performance metrics, the inverted generational distance (IGD) [38] and the coverage over the Pareto front (CPF) [39], were adopted to quantitatively evaluate the performance of the compared algorithms. The IGD can simultaneously evaluate the convergence and diversity of a solution set, while the CPF focuses on evaluating the population diversity of algorithms. For the IGD, the smaller the value, the better the evaluated algorithm performs. For the CPF, the larger the value, the better the obtained nondominated solutions distributed along the ideal Pareto fronts. To get statistical results, all algorithms were run 30 times independently for each test problem. 

### 4.3. Performance Comparison between LSMOEA-TM and Other Large-Scale MOEAs

Table 1 and Table 2 present the statistical results, i.e., the mean values and the standard deviations, of the IGD and CPF metrics of the five comparative algorithms on the 15 test problems with 100, 500, and 1000 decision variables obtained via 30 independent runs. The Wilcoxon rank sum test at a significance level of 0.05 was adopted to compare the performance of the algorithms, where the symbols “+”, “−”, and “=” indicate that the result is significantly better, significantly worse, and statistically similar to that obtained by LSMOEA-TM, respectively. Besides, the best results in Table 1 and Table 2 are shown in bold.

It can be observed from Table 1 and Table 2 that LSMOEA-TM achieved the best results for most of the test problems in terms of both IGD and CPF. This result indicated the effectiveness of the proposed LSMOEA-TM, which enhanced the optimization performance by adopting two alternative optimization methods. With the help of the two alternative optimization methods, LSMOEA-TM could achieve a balance between convergence and population diversity. For DTLZ1, LSMOEA-TM obtained results that were worse than LMEA in terms of both IGD and CPF. This may be because the group results of the decision variables obtained by LMEA based on the initial population were reasonable enough in most cases to solve DTLZ1. LSMOEA-TM spent some computations on adjusting the group results of the decision variables dynamically so that the number of computations consumed on the population’s evolution was smaller than that in LMEA. As computationally expensive problems, WFG2 and WFG3 needed a large number of computations to group the decision variables in LSMOEA-TM. Therefore, the number of computations left to update the population was reduced correspondingly. As shown in Table 1 and Table 2, the performance of LSMOEA-TM on WFG2 and WFG3 was worse than some of the comparative algorithms. For UF4 and UF7, the IGD values of LSMOEA-TM were slightly worse than the best results when the decision variables were lower than 500. However, LSMOEA-TM achieved the best statistical results on both IGD and CPF for UF4 and UF7 with 1000 decision variables. For BT1, BT2, BT3, and BT6, LSMOEA-TM obtained the best performance for both the IGD and CPF. This may demonstrate the effectiveness of LSMOEA-TM when handling large-scale multiobjective problems.

LSMOP [7] is a recently proposed test suite and can reflect many characteristics of real-world large-scale optimization problems. It can be observed from Table 3 and Table 4, where the best results are shown in bold that LSMOEA-TM achieved the best performance for LSMOP1, LSMOP4, LSMOP5, LSMOP8, and LSMOP9. For LSMOP7, the IGD and CPF values obtained by LSMOEA-TM were slightly worse than for FDV. For LSMOP2 and LSMOP3, the metric values of LSMOEA-TM had a certain gap with the best. That is because LSMOP2 is a unimodal multimodal mixed problem with partially separable decision variables, while LSMOP3 is a multimodal problem with partially separable and fully separable decision variables. Since LSMOEA-TM achieved efficient solutions to LSMOPs from the perspective of grouping decision variables, the effect of the grouping strategy may be affected when facing multimodal problems and problems with partially separable decision variables, thus resulting in a poor search performance. However, IM-MOEA/D decomposes the problem from the decision-variable space, and FDV uses a fuzzy search instead of a grouping strategy. Therefore, when facing this type of problem, the performance of LSMOEA-TM will be slightly worse than that of IM-MOEA/D and FDV. However, overall, LSMOEA-TM achieved the best performance on the LSMOP test suite.

## 5. Discussion

### 5.1. Investigation of the Bayesian-Based Parameter Adjusting Strategy

In this section, six problems, including DTLZ2, UF1, BT1, WFG2, LSMOP1, and LSMOP2, were selected to investigate the effectiveness of the proposed Bayesian-based parameter adjusting strategy. The number of decision variables was set to 100 for DTLZ2, UF1, BT1, and WFG2. For LSMOP1 and LSMOP2, the number of decision variables was 100×M, where M is the dimension of objective space. Since LSMOP1 and LSMOP are three-objective problems, the number of decision variables for these two problems was 300. For LSMOEA-TM without the Bayesian-based parameter-adjusting strategy, the parameters used in the grouping strategies, namely nSel and nPer, had the same values as in [16]. For LSMOEA-TM proposed in this paper, nSel and nPer were adjusted according to a Bayesian-based parameter-adjusting strategy to achieve the balance between grouping accuracy and computations. As shown in Figure 8, LSMOEA-TM could obtain the best IGD values faster than LSMOEA-TM without the Bayesian-based parameter-adjusting strategy for all test problems. This may indicate the effectiveness of the proposed Bayesian-based parameter-adjusting strategy, which could find the appropriate values for the parameters used in the grouping strategies. It can be observed from Figure 8 that LSMOEA-TM converged slower than LSMOEA-TM without the Bayesian-based parameter-adjusting strategy at the early stages for some problems, such as DTLZ2, UF1, and BT1. As the evolution progressed, more observation values were generated, and the probability surrogate model adopted in the Bayesian-based parameter-adjusting strategy was closer to the real loss function. Therefore, in the early stages of LSMOEA-TM, the Bayesian-based parameter-adjusting strategy may not obtain satisfactory values for nSel and nPer, thus leading to relatively poor IGD values. However, as shown in Figure 8, the Bayesian-based parameter-adjusting strategy could help LSMOEA-TM find appropriate parameter values in the middle and late stages and finally achieve a better performance. 

### 5.2. Investigation of the Scalability of LSMOEA-TM

To further investigate the scalability of LSMOEA-TM, the test problems with more decision variables were adopted in this section. Figure 9 presents the average IGD values obtained by LSMOEA-TM over 30 independent runs for DTLZ1, DTLZ3, UF5, UF10, WFG2, and WFG6, in which the number of decision variables ranged from 100 to 4000. As shown in Figure 9, the IGD values fluctuated with the number of decision variables increasing. For all six test problems, there was no significant degradation in the performance of LSMOEA-TM when the number of decision variables ranged from 100 to 4000. The experiment results demonstrated that LSMOEA-TM had a stable performance for large-scale MOPs with the different decision variables.

### 5.3. Investigation of the Computational Efficiency of LSMOEA-TM

To save space, Table 5 presents the runtime of five algorithms on some selected problems. For a comprehensive comparison, Table 6 gives the average runtime of five algorithms on all problems of each test suite. It should be noted that the number of decision variables in the LSMOP suite was set to a fixed constant, i.e., 300, so the runtime of LSMOP was obviously less than that of the other test suites for all five algorithms.

Because of the simplicity of the procedure, MOEA/D2 spent the least time among all algorithms. It can be observed from both Table 5 and Table 6 that LSMOEAs based on grouping strategies, i.e., LMEA and the proposed LSMOEA-TM, spent less time than IM-MOEA/D and FDV. For the DTLZ, UF, and LSMOP test suites, LSMOEA-TM cost less runtime than LMEA, IM-MOEA/D, and FDV. This was because the proposed Bayesian-based parameter-adjustment strategy helped LSMOEA-TM reduce the computational cost by estimating the appropriate parameters (nSel and nPer) in its grouping strategies. However, LSMOEA-TM took longer than LMEA when solving computationally expensive problems, such as the WFG test problems. This was because the calculation costs of the HV metric, which was used to control the alternation of the convergence-oriented and diversity-oriented stages in LSMOEA-TM, for the WFG test suite were significantly higher than those for the other test suites. For the BT test suite, LSMOEA-TM spent more time in some cases. This was because LSMOEA-TM had to execute more grouping strategies to achieve the balance of convergence and diversity since it was difficult to maintain the population diversity for the BT test suite.

## 6. Conclusions

This paper proposed a large-scale multiobjective optimization algorithm with two alternative optimization methods (LSMOEA-TM) to solve LSMOPs. In the two alternative optimization methods, two grouping strategies, namely, the convergence-related and the diversity-related grouping strategies, were used to divide decision variables and get stable grouping results. Specifically, LSMOEA-TM utilized the grouping strategy chosen from the two grouping strategies mentioned above according to the performance of the evolved population to balance convergence and population diversity. Furthermore, this paper introduced a Bayesian-based parameter-adjusting strategy to reduce computational costs by optimizing the parameters in the proposed two alternative optimization methods. The proposed LSMOEA-TM was compared with four state-of-the-art large-scale optimization algorithms on benchmark large-scale test problems in the experimental section. The statistical results demonstrated that LSMOEA-TM performed best for most of the test problems. This indicated the effectiveness of the proposed two alternative optimization methods in solving LSMOPs. Moreover, the results in Section 4.2 showed that the proposed Bayesian-based parameter-adjusting strategy could reduce computational costs and improve the search efficiency of LSMOEA-TM. In addition, Figure 9 demonstrated that there was no significant degradation in the performance of LSMOEA-TM when extended to LSMOPs with more decision variables.

As described in Section 3.2, LSMOEA-TM used the grouping strategy chosen from the convergence-related and diversity-related grouping strategies according to the performance of the evolved population, which was evaluated by the widely adopted HV metric. However, the calculation costs of HV will increase exponentially with the increase of the dimension of the objective space, so how to design an efficient evaluation method is left for future work. In addition, how to improve the performance of LSMOEA-TM in solving complex LSMOPs, such as problems with partially separable decision variables and problems with multimodal characteristics, is also a topic for future work.

## Figures and Tables

**Figure 1 entropy-25-00561-f001:**
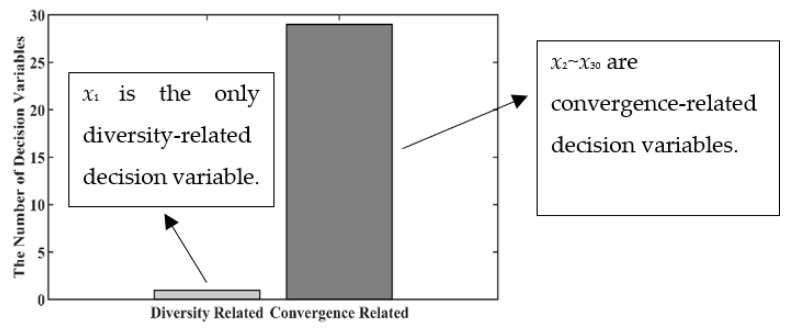
Grouping Results of BT6 by LMEA.

**Figure 2 entropy-25-00561-f002:**
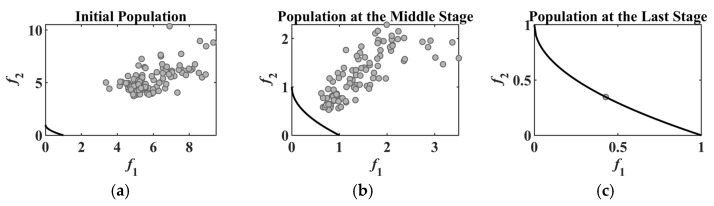
Evolution Process of BT6 by LMEA. (**a**) evolution starts; (**b**) evolution processes half; (**c**) evolution ends.

**Figure 3 entropy-25-00561-f003:**
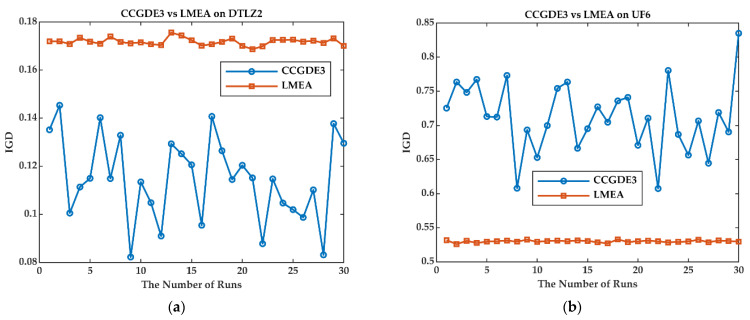
Illustration of the Stability of CCGDE3 and LMEA. (**a**) on DTLZ2; (**b**) on UF6.

**Figure 4 entropy-25-00561-f004:**
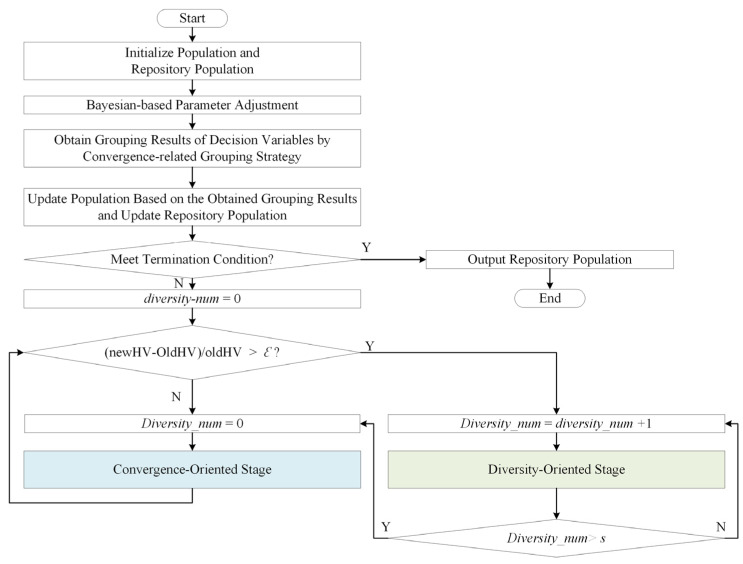
Framework of LSMOEA-TM.

**Figure 5 entropy-25-00561-f005:**
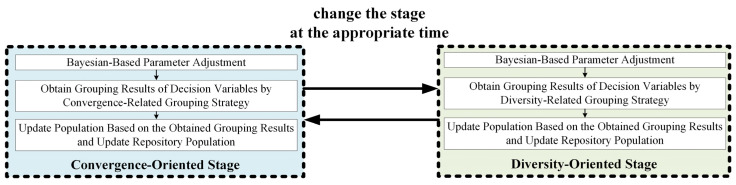
Illustration of Convergence-Oriented and Diversity-Oriented Stages.

**Figure 6 entropy-25-00561-f006:**
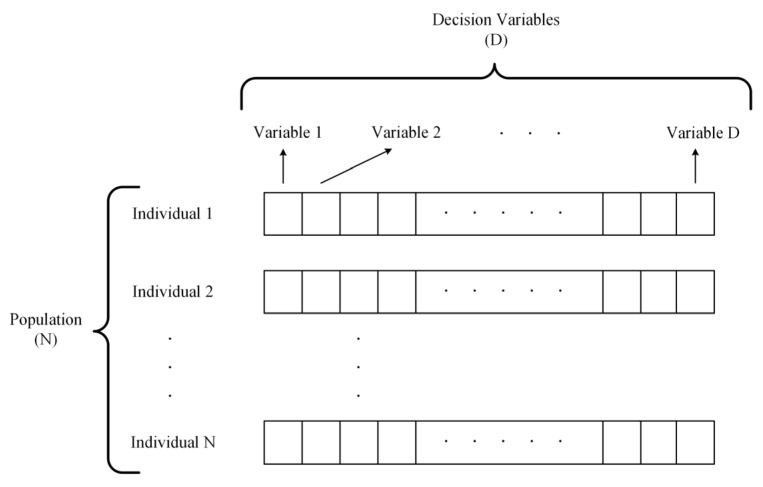
Illustration of the Structure of the Population.

**Figure 7 entropy-25-00561-f007:**
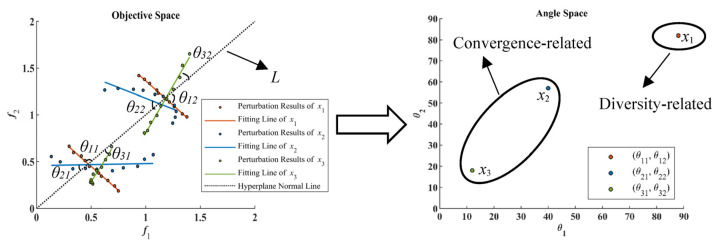
Illustration of Perturbation Results for Decision Variables.

**Figure 8 entropy-25-00561-f008:**
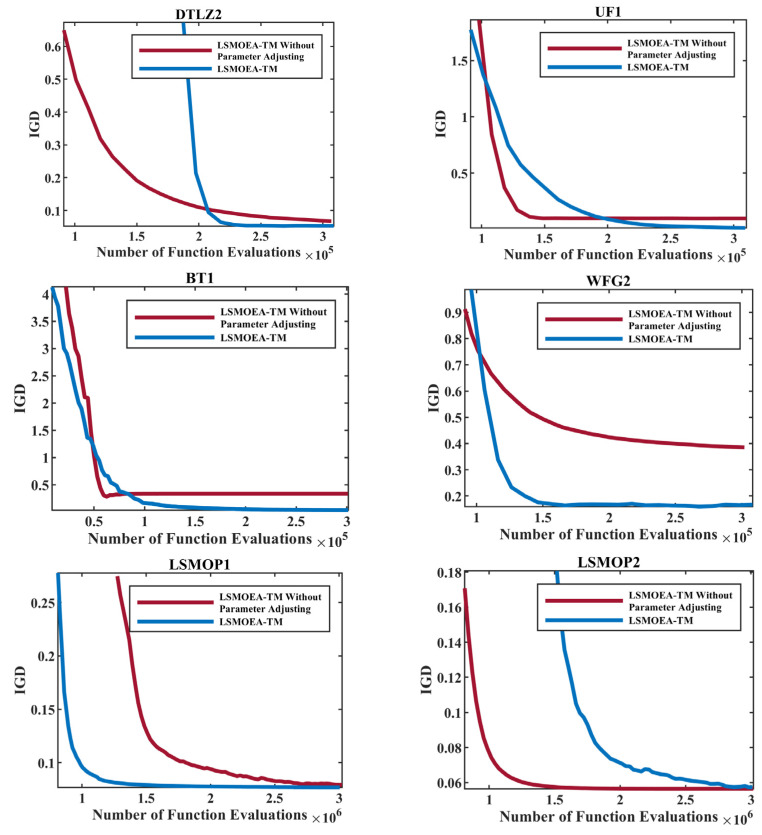
Comparison of LSMOEA-TM with and without Bayesian-based Parameter Adjusting Strategy.

**Figure 9 entropy-25-00561-f009:**
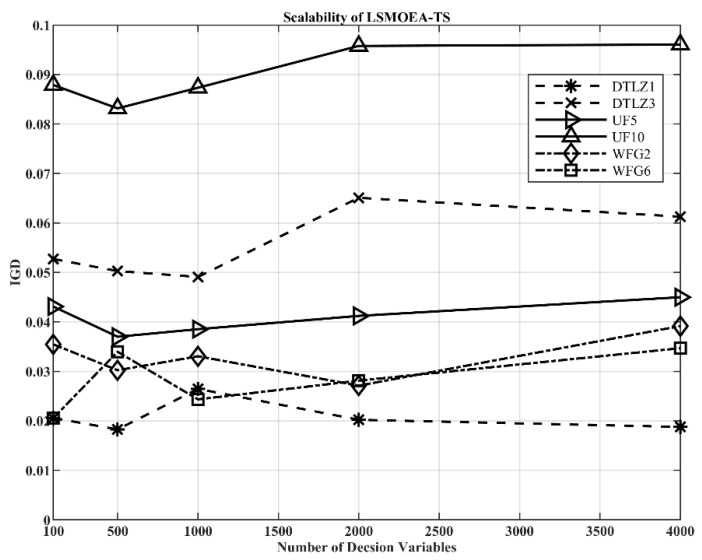
IGD Metric Values of LSMOEA-TM on Six Problems with Different Numbers of Decision Variables, Averaged over 30 Runs.

**Table 1 entropy-25-00561-t001:** IGD values of Five Algorithms on DTLZ, UF, WFG, and BT Test Suites.

Problem	D	MOEA/D2	LMEA	IM-MOEA/D	FDV	LSMOEA-TM
DTLZ1	100	1.17 × 10^3^ (2.45 × 10^2^) −	**2.05 × 10^−2^ (1.42 × 10^−6^) +**	3.16 × 10^−1^ (1.05 × 10^−1^) −	2.05 × 10^−2^ (8.72 × 10^−6^) +	2.09 × 10^−2^ (2.53 × 10^−4^)
	500	3.03 × 10^3^ (3.54 × 10^2^) −	**2.05 × 10^−2^ (1.59 × 10^−6^) +**	5.34 × 10^0^ (8.06 × 10^−1^) −	2.12 × 10^−2^ (2.20 × 10^−4^) =	2.11 × 10^−2^ (3.96 × 10^−4^)
	1000	4.67 × 10^3^ (6.89 × 10^2^) −	**2.05 × 10^−2^ (1.84 × 10^−6^) +**	1.65 × 10^1^ (1.57 × 10^0^) −	2.87 × 10^−2^ (1.65 × 10^−3^) −	2.12 × 10^−2^ (3.01 × 10^−4^)
DTLZ2	100	1.90 × 10^0^ (6.12 × 10^−1^) −	5.44 × 10^−2^ (3.85 × 10^−6^) −	5.92 × 10^−2^ (1.17 × 10^−9^) −	5.44 × 10^−2^ (1.82 × 10^−6^) −	**5.33 × 10^−2^ (4.29 × 10^−4^)**
	500	4.42 × 10^0^ (8.40 × 10^−1^) −	5.44 × 10^−2^ (5.19 × 10^−6^) −	5.92 × 10^−2^ (6.74 × 10^−9^) −	5.44 × 10^−2^ (6.38 × 10^−8^) −	**5.38 × 10^−2^ (5.64 × 10^−4^)**
	1000	7.74 × 10^0^ (1.71 × 10^0^) −	5.44 × 10^−2^ (4.25 × 10^−6^) −	5.92 × 10^−2^ (3.75 × 10^−8^) −	5.44 × 10^−2^ (2.73 × 10^−8^) −	**5.39 × 10^−2^ (4.68 × 10^−4^)**
DTLZ3	100	3.37 × 10^3^ (7.76 × 10^2^) −	5.44 × 10^−2^ (3.69 × 10^−6^) −	8.61 × 10^−1^ (3.98 × 10^−1^) −	5.45 × 10^−2^ (1.86 × 10^−5^) −	**5.34 × 10^−2^ (5.10 × 10^−4^)**
	500	7.88 × 10^3^ (1.21 × 10^3^) −	5.44 × 10^−2^ (5.79 × 10^−6^) −	1.41 × 10^1^ (2.29 × 10^0^) −	5.66 × 10^−2^ (8.20 × 10^−4^) −	**5.40 × 10^−2^ (6.39 × 10^−4^)**
	1000	1.32 × 10^4^ (2.19 × 10^3^) −	**5.44 × 10^−2^ (4.94 × 10^−6^) =**	4.60 × 10^1^ (6.83 × 10^0^) −	7.97 × 10^−2^ (5.60 × 10^−3^) −	5.46 × 10^−2^ (7.88 × 10^−4^)
DTLZ7	100	3.21 × 10^0^ (3.66 × 10^−1^) −	2.91 × 10^−1^ (1.81 × 10^−1^) −	1.24 × 10^−1^ (1.05 × 10^−16^) −	7.76 × 10^−2^ (3.20 × 10^−3^) −	**5.89 × 10^−2^ (1.33 × 10^−3^)**
	500	3.72 × 10^0^ (2.29 × 10^−1^) −	2.85 × 10^−1^ (1.83 × 10^−1^) −	1.24 × 10^−1^ (5.70 × 10^−13^) −	7.89 × 10^−2^ (3.01 × 10^−3^) −	**5.90 × 10^−2^ (1.21 × 10^−3^)**
	1000	3.97 × 10^0^ (1.91 × 10^−1^) −	2.46 × 10^−1^ (1.85 × 10^−1^) −	1.24 × 10^−1^ (1.89 × 10^−7^) −	7.85 × 10^−2^ (3.62 × 10^−3^) −	**5.94 × 10^−2^ (1.21 × 10^−3^)**
UF1	100	5.92 × 10^−1^ (7.68 × 10^−2^) −	5.92 × 10^−1^ (7.68 × 10^−2^) −	8.30 × 10^−2^ (9.00 × 10^−3^) −	8.06 × 10^−3^ (3.34 × 10^−3^) −	**3.73 × 10^−3^ (1.56 × 10^−8^)**
	500	6.92 × 10^−1^ (7.72 × 10^−2^) −	6.92 × 10^−1^ (7.72 × 10^−2^) −	9.33 × 10^−2^ (1.02 × 10^−2^) −	8.12 × 10^−3^ (2.78 × 10^−3^) −	**3.73 × 10^−3^ (4.78 × 10^−8^)**
	1000	7.55 × 10^−1^ (7.47 × 10^−2^) −	7.55 × 10^−1^ (7.47 × 10^−2^) −	1.01 × 10^−1^ (1.19 × 10^−2^) −	8.22 × 10^−3^ (2.41 × 10^−3^) −	**3.73 × 10^−3^ (9.09 × 10^−8^)**
UF2	100	2.18 × 10^−1^ (3.94 × 10^−2^) −	2.18 × 10^−1^ (3.94 × 10^−2^) −	4.22 × 10^−2^ (2.09 × 10^−2^) −	7.96 × 10^−3^ (1.07 × 10^−3^) −	**3.73 × 10^−3^ (1.17 × 10^−9^)**
	500	2.88 × 10^−1^ (5.56 × 10^−2^) −	2.88 × 10^−1^ (5.56 × 10^−2^) −	5.06 × 10^−2^ (2.08 × 10^−2^) −	8.37 × 10^−3^ (1.11 × 10^−3^) −	**3.73 × 10^−3^ (4.36 × 10^−9^)**
	1000	3.23 × 10^−1^ (5.93 × 10^−2^) −	3.23 × 10^−1^ (5.93 × 10^−2^) −	5.69 × 10^−2^ (1.75 × 10^−2^) −	9.15 × 10^−3^ (9.22 × 10^−4^) −	**3.73 × 10^−3^ (1.09 × 10^−8^)**
UF4	100	1.07 × 10^−1^ (4.09 × 10^−3^) −	5.59 × 10^−2^ (3.01 × 10^−3^) −	4.21 × 10^−2^ (1.97 × 10^−3^) −	**1.09 × 10^−2^ (1.11 × 10^−3^) +**	2.07 × 10^−2^ (1.64 × 10^−4^)
	500	1.33 × 10^−1^ (5.55 × 10^−3^) −	6.27 × 10^−2^ (5.02 × 10^−3^) −	5.12 × 10^−2^ (2.23 × 10^−3^) −	**1.89 × 10^−2^ (1.16 × 10^−3^) +**	2.32 × 10^−2^ (1.05 × 10^−4^)
	1000	1.45 × 10^−1^ (3.63 × 10^−3^) −	6.78 × 10^−2^ (4.50 × 10^−3^) −	5.63 × 10^−2^ (1.86 × 10^−3^) −	2.59 × 10^−2^ (4.84 × 10^−4^) −	**2.43 × 10^−2^ (8.24 × 10^−5^)**
UF7	100	6.22 × 10^−1^ (1.14 × 10^−1^) −	9.72 × 10^−2^ (2.08 × 10^−1^) −	8.02 × 10^−2^ (1.03 × 10^−1^) −	**3.24 × 10^−2^ (3.94 × 10^−2^) +**	5.97 × 10^−2^ (5.79 × 10^−7^)
	500	8.01 × 10^−1^ (8.70 × 10^−2^) −	2.23 × 10^−1^ (3.12 × 10^−1^) −	8.48 × 10^−2^ (9.00 × 10^−2^) −	**4.77 × 10^−2^ (4.55 × 10^−2^) +**	5.97 × 10^−2^ (1.42 × 10^−6^)
	1000	8.37 × 10^−1^ (9.75 × 10^−2^) −	2.52 × 10^−1^ (3.22 × 10^−1^) =	**5.67 × 10^−2^ (1.34 × 10^−2^) +**	8.07 × 10^−2^ (6.75 × 10^−2^) =	5.97 × 10^−2^ (2.55 × 10^−6^)
WFG1	100	2.24 × 10^0^ (7.82 × 10^−2^) −	1.38 × 10^0^ (1.34 × 10^−1^) −	3.11 × 10^−1^ (2.42 × 10^−2^) +	**1.41 × 10^−1^ (4.28 × 10^−4^) +**	6.32 × 10^−1^ (8.79 × 10^−2^)
	500	2.25 × 10^0^ (7.21 × 10^−2^) −	1.50 × 10^0^ (1.18 × 10^−1^) −	2.74 × 10^−1^ (3.68 × 10^−2^) +	**1.41 × 10^−1^ (7.56 × 10^−5^) +**	8.13 × 10^−1^ (7.58 × 10^−2^)
	1000	2.24 × 10^0^ (7.73 × 10^−2^) −	1.52 × 10^0^ (9.55 × 10^−2^) −	2.87 × 10^−1^ (1.96 × 10^−1^) +	**1.41 × 10^−1^ (2.15 × 10^−5^) +**	7.77 × 10^−1^ (9.23 × 10^−2^)
WFG2	100	4.65 × 10^−1^ (5.75 × 10^−4^) −	5.71 × 10^−1^ (4.84 × 10^−2^) −	1.87 × 10^−1^ (4.60 × 10^−3^) −	1.65 × 10^−1^ (1.01 × 10^−3^) −	**1.65 × 10^−1^ (5.24 × 10^−3^)**
	500	4.65 × 10^−1^ (5.97 × 10^−4^) −	5.68 × 10^−1^ (3.79 × 10^−2^) −	2.14 × 10^−1^ (6.55 × 10^−3^) −	1.74 × 10^−1^ (3.12 × 10^−3^) −	**1.74 × 10^−1^ (5.98 × 10^−2^)**
	1000	4.65 × 10^−1^ (4.53 × 10^−4^) −	5.85 × 10^−1^ (3.50 × 10^−2^) −	2.24 × 10^−1^ (6.22 × 10^−3^) −	1.83 × 10^−1^ (4.43 × 10^−3^) −	**1.63 × 10^−1^ (3.33 × 10^−3^)**
WFG3	100	1.08 × 10^−1^ (2.65 × 10^−2^) −	6.06 × 10^−1^ (4.08 × 10^−2^) −	2.33 × 10^−1^ (1.72 × 10^−2^) −	6.76 × 10^−2^ (6.12 × 10^−3^) −	**3.06 × 10^−2^ (4.74 × 10^−3^)**
	500	1.15 × 10^−1^ (1.88 × 10^−2^) −	6.62 × 10^−1^ (4.51 × 10^−2^) −	2.60 × 10^−1^ (2.05 × 10^−2^) −	9.22 × 10^−2^ (6.72 × 10^−3^) −	**3.14 × 10^−2^ (3.90 × 10^−3^)**
	1000	1.19 × 10^−1^ (1.62 × 10^−2^) −	6.66 × 10^−1^ (4.51 × 10^−2^) −	2.56 × 10^−1^ (2.02 × 10^−2^) −	1.36 × 10^−1^ (2.37 × 10^−2^) −	**3.19 × 10^−2^ (4.13 × 10^−3^)**
BT1	100	1.09 × 10^1^ (8.92 × 10^−1^) −	9.25 × 10^0^ (9.19 × 10^−1^) −	1.74 × 10^0^ (3.03 × 10^−1^) −	1.83 × 10^0^ (2.16 × 10^−1^) −	**3.77 × 10^−2^ (8.90 × 10^−3^)**
	500	2.54 × 10^1^ (3.08 × 10^0^) −	1.79 × 10^1^ (1.53 × 10^0^) −	3.86 × 10^0^ (5.68 × 10^−1^) −	4.42 × 10^0^ (3.39 × 10^−1^) −	**7.97 × 10^−2^ (1.28 × 10^−2^)**
	1000	3.96 × 10^1^ (4.46 × 10^0^) −	2.85 × 10^1^ (2.46 × 10^0^) −	8.49 × 10^0^ (8.65 × 10^−1^) −	9.39 × 10^0^ (4.05 × 10^−1^) −	**1.33 × 10^−1^ (2.25 × 10^−2^)**
BT2	100	8.42 × 10^0^ (6.39 × 10^−1^) −	1.85 × 10^0^ (9.82 × 10^−2^) −	7.84 × 10^−1^ (4.69 × 10^−2^) −	8.18 × 10^−1^ (1.96 × 10^−2^) −	**3.45 × 10^−1^ (4.32 × 10^−2^)**
	500	1.91 × 10^1^ (1.29 × 10^0^) −	3.95 × 10^0^ (2.02 × 10^−1^) −	1.65 × 10^0^ (4.89 × 10^−2^) −	1.77 × 10^0^ (3.10 × 10^−2^) −	**7.39 × 10^−1^ (5.64 × 10^−2^)**
	1000	3.06 × 10^1^ (1.58 × 10^0^) −	6.40 × 10^0^ (2.18 × 10^−1^) −	2.66 × 10^0^ (7.64 × 10^−2^) −	3.21 × 10^0^ (4.68 × 10^−2^) −	**1.22 × 10^0^ (8.36 × 10^−2^)**
BT3	100	1.12 × 10^1^ (1.36 × 10^0^) −	3.23 × 10^0^ (9.68 × 10^−1^) −	2.33 × 10^−1^ (6.59 × 10^−2^) −	8.81 × 10^−1^ (9.40 × 10^−2^) −	**1.05 × 10^−2^ (2.82 × 10^−3^)**
	500	2.42 × 10^1^ (3.08 × 10^0^) −	6.69 × 10^0^ (1.30 × 10^0^) −	2.71 × 10^−1^ (8.65 × 10^−2^) −	1.90 × 10^0^ (1.77 × 10^−1^) −	**1.83 × 10^−2^ (4.97 × 10^−3^)**
	1000	3.86 × 10^1^ (4.71 × 10^0^) −	1.23 × 10^1^ (1.70 × 10^0^) −	5.06 × 10^−1^ (1.02 × 10^−1^) −	3.85 × 10^0^ (3.29 × 10^−1^) −	**3.05 × 10^−2^ (5.04 × 10^−3^)**
BT6	100	1.10 × 10^1^ (1.05 × 10^0^) −	9.12 × 10^0^ (8.70 × 10^−1^) −	1.47 × 10^0^ (3.43 × 10^−1^) −	1.95 × 10^0^ (1.92 × 10^−1^) −	**2.61 × 10^−2^ (1.01 × 10^−2^)**
	500	2.52 × 10^1^ (3.16 × 10^0^) −	1.80 × 10^1^ (1.88 × 10^0^) −	3.77 × 10^0^ (5.81 × 10^−1^) −	4.65 × 10^0^ (2.31 × 10^−1^) −	**5.29 × 10^−2^ (8.05 × 10^−3^)**
	1000	3.94 × 10^1^ (4.66 × 10^0^) −	2.82 × 10^1^ (3.08 × 10^0^) −	7.98 × 10^0^ (6.22 × 10^−1^) −	9.53 × 10^0^ (4.77 × 10^−1^) −	**8.38 × 10^−2^ (1.36 × 10^−2^)**
+/−/=		0/45/0	3/40/2	4/41/0	8/36/1	

**Table 2 entropy-25-00561-t002:** CPF Values of Five Algorithms on DTLZ, UF, WFG, and BT Test Suites.

Problem	D	MOEA/D2	LMEA	IM-MOEA/D	FDV	LSMOEA-TM
DTLZ1	100	0.00 × 10^0^ (0.00 × 10^0^) −	**8.41 × 10^−1^ (3.51 × 10^−5^) +**	1.56 × 10^−1^ (1.39 × 10^−1^) −	8.41 × 10^−1^ (1.06 × 10^−4^) +	8.40 × 10^−1^ (5.22 × 10^−4^)
	500	0.00 × 10^0^ (0.00 × 10^0^) −	**8.41 × 10^−1^ (3.09 × 10^−5^) +**	0.00 × 10^0^ (0.00 × 10^0^) −	8.36 × 10^−1^ (1.25 × 10^−3^) −	8.39 × 10^−1^ (8.68 × 10^−4^)
	1000	0.00 × 10^0^ (0.00 × 10^0^) −	**8.41 × 10^−1^ (2.27 × 10^−5^) +**	0.00 × 10^0^ (0.00 × 10^0^) −	8.11 × 10^−1^ (4.36 × 10^−3^) −	8.37 × 10^−1^ (8.24 × 10^−4^)
DTLZ2	100	0.00 × 10^0^ (0.00 × 10^0^) −	5.59 × 10^−1^ (3.46 × 10^−5^) −	5.39 × 10^−1^ (3.94 × 10^−8^) −	5.59 × 10^−1^ (1.70 × 10^−5^) −	**5.61 × 10^−1^ (5.84 × 10^−4^)**
	500	0.00 × 10^0^ (0.00 × 10^0^) −	5.59 × 10^−1^ (4.56 × 10^−5^) −	5.39 × 10^−1^ (3.09 × 10^−7^) −	5.59 × 10^−1^ (1.83 × 10^−7^) −	**5.61 × 10^−1^ (7.25 × 10^−4^)**
	1000	0.00 × 10^0^ (0.00 × 10^0^) −	5.59 × 10^−1^ (4.19 × 10^−5^) −	5.39 × 10^−1^ (1.41 × 10^−6^) −	5.59 × 10^−1^ (7.20 × 10^−8^) −	**5.60 × 10^−1^ (8.03 × 10^−4^)**
DTLZ3	100	0.00 × 10^0^ (0.00 × 10^0^) −	5.59 × 10^−1^ (5.74 × 10_−5_) −	4.24 × 10^−2^ (4.42 × 10^−2^) −	5.58 × 10^−1^ (4.27 × 10^−4^) −	**5.61 × 10^−1^ (9.12 × 10^−4^)**
	500	0.00 × 10^0^ (0.00 × 10^0^) −	5.59 × 10^−1^ (5.04 × 10^−5^) −	0.00 × 10^0^ (0.00 × 10^0^) −	5.42 × 10^−1^ (3.66 × 10^−3^) −	**5.59 × 10^−1^ (6.82 × 10^−4^)**
	1000	0.00 × 10^0^ (0.00 × 10^0^) −	**5.59 × 10^−1^ (4.40 × 10^−5^) +**	0.00 × 10^0^ (0.00 × 10^0^) −	4.84 × 10^−1^ (1.14 × 10^−2^) −	5.54 × 10^−1^ (1.65 × 10^−3^)
DTLZ7	100	0.00 × 10^0^ (0.00 × 10^0^) −	2.42 × 10^−1^ (1.58 × 10^−2^) −	2.60 × 10^−1^ (1.01 × 10^−16^) −	2.67 × 10^−1^ (1.72 × 10^−3^) −	**2.79 × 10^−1^ (6.07 × 10^−4^)**
	500	0.00 × 10^0^ (0.00 × 10^0^) −	2.42 × 10^−1^ (1.62 × 10^−2^) −	2.60 × 10^−1^ (1.03 × 10^−14^) −	2.68 × 10^−1^ (1.73 × 10^−3^) −	**2.79 × 10^−1^ (6.12 × 10^−4^)**
	1000	0.00 × 10^0^ (0.00 × 10^0^) −	2.46 × 10^−1^ (1.65 × 10^−2^) −	2.60 × 10^−1^ (4.84 × 10^−9^) −	2.69 × 10^−1^ (1.55 × 10^−3^) −	**2.79 × 10^−1^ (6.79 × 10^−4^)**
UF1	100	1.02 × 10^−1^ (4.38 × 10^−2^) −	7.16 × 10^−1^ (9.80 × 10^−3^) −	7.14 × 10^−1^ (3.98 × 10^−3^) −	6.23 × 10^−1^ (1.60 × 10^−2^) −	**7.20 × 10^−1^ (6.99 × 10^−7^)**
	500	5.78 × 10^−2^ (3.52 × 10^−2^) −	7.14 × 10^−1^ (1.27 × 10^−2^) −	7.14 × 10^−1^ (3.07 × 10^−3^) −	6.12 × 10^−1^ (1.84 × 10^−2^) −	**7.20 × 10^−1^ (1.51 × 10^−6^)**
	1000	3.74 × 10^−2^ (2.01 × 10^−2^) −	6.87 × 10^−1^ (9.42 × 10^−2^) −	7.14 × 10^−1^ (2.42 × 10^−3^) −	6.00 × 10^−1^ (1.70 × 10^−2^) −	**7.20 × 10^−1^ (1.80 × 10^−6^)**
UF2	100	4.48 × 10^−1^ (4.00 × 10^−2^) −	7.17 × 10^−1^ (6.27 × 10^−4^) −	7.14 × 10^−1^ (1.22 × 10^−3^) −	6.86 × 10^−1^ (1.12 × 10^−2^) −	**7.20 × 10^−1^ (3.93 × 10^−8^)**
	500	3.77 × 10^−1^ (5.18 × 10^−2^) −	7.16 × 10^−1^ (6.13 × 10^−4^) −	7.13 × 10^−1^ (1.47 × 10^−3^) −	6.76 × 10^−1^ (1.17 × 10^−2^) −	**7.20 × 10^−1^ (1.07 × 10^−7^)**
	1000	3.44 × 10^−1^ (5.36 × 10^−2^) −	7.15 × 10^−1^ (8.32 × 10^−4^) −	7.11 × 10^−1^ (1.26 × 10^−3^) −	6.68 × 10^−1^ (1.10 × 10^−2^) −	**7.20 × 10^−1^ (2.16 × 10^−7^)**
UF4	100	3.00 × 10^−1^ (5.28 × 10^−3^) −	3.67 × 10^−1^ (4.10 × 10^−3^) −	**4.35 × 10^−1^ (1.50 × 10^−3^) +**	3.89 × 10^−1^ (2.12 × 10^−3^) −	4.17 × 10^−1^ (2.61 × 10^−4^)
	500	2.66 × 10^−1^ (6.65 × 10^−3^) −	3.58 × 10^−1^ (6.78 × 10^−3^) −	**4.21 × 10^−1^ (1.22 × 10^−3^) +**	3.77 × 10^−1^ (2.90 × 10^−3^) −	4.13 × 10^−1^ (1.87 × 10^−4^)
	1000	2.52 × 10^−1^ (3.97 × 10^−3^) −	3.51 × 10^−1^ (6.02 × 10^−3^) −	4.09 × 10^−1^ (1.47 × 10^−3^) −	3.68 × 10^−1^ (2.61 × 10^−3^) −	**4.12 × 10^−1^ (1.44 × 10^−4^)**
UF7	100	3.85 × 10^−2^ (3.73 × 10^−2^) −	5.08 × 10^−1^ (1.45 × 10^−1^) −	**5.51 × 10^−1^ (3.63 × 10^−2^) +**	5.05 × 10^−1^ (7.37 × 10^−2^) −	5.16 × 10^−1^ (1.46 × 10^−6^)
	500	3.65 × 10^−3^ (6.82 × 10^−3^) −	4.20 × 10^−1^ (2.14 × 10^−1^) −	**5.34 × 10^−1^ (3.96 × 10^−2^) +**	4.97 × 10^−1^ (6.72 × 10^−2^) −	5.16 × 10^−1^ (2.45 × 10^−6^)
	1000	1.82 × 10^−3^ (3.67 × 10^−3^) −	3.98 × 10^−1^ (2.20 × 10^−1^) =	5.51 × 10^−1^ (3.63 × 10^−2^) +	5.15 × 10^−1^ (1.57 × 10^−2^) −	**5.16 × 10^−1^ (4.55 × 10^−6^)**
WFG1	100	3.09 × 10^−4^ (1.69 × 10^−3^) −	3.62 × 10^−1^ (5.77 × 10^−2^) −	5.34 × 10^−1^ (3.96 × 10^−2^) +	**9.44 × 10^−1^ (1.36 × 10^−4^) +**	8.34 × 10^−1^ (1.96 × 10^−2^)
	500	0.00 × 10^0^ (0.00 × 10^0^) −	3.03 × 10^−1^ (3.93 × 10^−2^) −	5.51 × 10^−1^ (3.63 × 10^−2^) +	**9.44 × 10^−1^ (4.67 × 10^−5^) +**	7.42 × 10^−1^ (2.91 × 10^−2^)
	1000	4.83 × 10^−4^ (2.65 × 10^−3^) −	2.94 × 10^−1^ (3.38 × 10^−2^) −	5.34 × 10^−1^ (3.96 × 10^−2^) +	**9.44 × 10^−1^ (2.87 × 10^−5^) +**	7.67 × 10^−1^ (2.67 × 10^−2^)
WFG2	100	8.64 × 10^−1^ (3.93 × 10^−3^) −	6.77 × 10^−1^ (1.77 × 10^−2^) −	5.51 × 10^−1^ (3.63 × 10^−2^) +	9.23 × 10^−1^ (2.24 × 10^−3^) −	**9.26 × 10^−1^ (2.55 × 10^−3^)**
	500	8.60 × 10^−1^ (2.81 × 10^−3^) −	6.71 × 10^−1^ (1.27 × 10^−2^) −	5.34 × 10^−1^ (3.96 × 10^−2^) +	9.00 × 10^−1^ (5.24 × 10^−3^) −	**9.21 × 10^−1^ (2.62 × 10^−2^)**
	1000	8.60 × 10^−1^ (2.74 × 10^−3^) −	6.61 × 10^−1^ (1.29 × 10^−2^) −	5.51 × 10^−1^ (3.63 × 10^−2^) +	8.87 × 10^−1^ (5.44 × 10^−3^) −	**9.24 × 10^−1^ (2.93 × 10^−3^)**
WFG3	100	3.66 × 10^−1^ (1.33 × 10^−2^) −	1.73 × 10^−1^ (9.82 × 10^−3^) −	5.34 × 10^−1^ (3.96 × 10^−2^) +	3.96 × 10^−1^ (2.14 × 10^−3^) −	**4.13 × 10^−1^ (3.08 × 10^−3^)**
	500	3.62 × 10^−1^ (9.48 × 10^−3^) −	1.53 × 10^−1^ (1.35 × 10^−2^) −	5.51 × 10^−1^ (3.63 × 10^−2^) +	3.84 × 10^−1^ (2.91 × 10^−3^) −	**4.11 × 10^−1^ (2.50 × 10^−3^)**
	1000	3.60 × 10^−1^ (7.88 × 10^−3^) −	1.51 × 10^−1^ (1.02 × 10^−2^) −	5.34 × 10^−1^ (3.96 × 10^−2^) +	3.62 × 10^−1^ (1.14 × 10^−2^) −	**4.10 × 10^−1^ (2.66 × 10^−3^)**
BT1	100	0.00 × 10^0^ (0.00 × 10^0^) −	0.00 × 10^0^ (0.00 × 10^0^) −	5.51 × 10^−1^ (3.63 × 10^−2^) +	0.00 × 10^0^ (0.00 × 10^0^) −	**6.71 × 10^−1^ (1.14 × 10^−2^)**
	500	0.00 × 10^0^ (0.00 × 10^0^) −	0.00 × 10^0^ (0.00 × 10^0^) −	5.34 × 10^−1^ (3.96 × 10^−2^) +	0.00 × 10^0^ (0.00 × 10^0^) −	**6.18 × 10^−1^ (1.61 × 10^−2^)**
	1000	0.00 × 10^0^ (0.00 × 10^0^) −	0.00 × 10^0^ (0.00 × 10^0^) −	5.51 × 10^−1^ (3.63 × 10^−2^) +	0.00 × 10^0^ (0.00 × 10^0^) −	**5.53 × 10^−1^ (2.72 × 10^−2^)**
BT2	100	0.00 × 10^0^ (0.00 × 10^0^) −	0.00 × 10^0^ (0.00 × 10^0^) −	5.34 × 10^−1^ (3.96 × 10^−2^) +	1.34 × 10^−2^ (4.55 × 10^−3^) −	**3.24 × 10^−1^ (4.09 × 10^−2^)**
	500	0.00 × 10^0^ (0.00 × 10^0^) −	0.00 × 10^0^ (0.00 × 10^0^) −	5.51 × 10^−1^ (3.63 × 10^−2^) +	0.00 × 10^0^ (0.00 × 10^0^) −	**5.49 × 10^−2^ (2.24 × 10^−2^)**
	1000	0.00 × 10^0^ (0.00 × 10^0^) −	0.00 × 10^0^ (0.00 × 10^0^) −	5.34 × 10^−1^ (3.96 × 10^−2^) +	0.00 × 10^0^ (0.00 × 10^0^) −	**3.72 × 10^−1^ (9.24 × 10^−3^)**
BT3	100	0.00 × 10^0^ (0.00 × 10^0^) −	0.00 × 10^0^ (0.00 × 10^0^) −	5.51 × 10^−1^ (3.63 × 10^−2^) +	1.45 × 10^−3^ (3.99 × 10^−3^) −	**7.06 × 10^−1^ (4.61 × 10^−3^)**
	500	0.00 × 10^0^ (0.00 × 10^0^) −	0.00 × 10^0^ (0.00 × 10^0^) −	5.34 × 10^−1^ (3.96 × 10^−2^) +	0.00 × 10^0^ (0.00 × 10^0^) −	**6.96 × 10^−1^ (7.10 × 10^−3^)**
	1000	0.00 × 10^0^ (0.00 × 10^0^) −	0.00 × 10^0^ (0.00 × 10^0^) −	5.51 × 10^−1^ (3.63 × 10^−2^) +	0.00 × 10^0^ (0.00 × 10^0^) −	**6.79 × 10^−1^ (6.91 × 10^−3^)**
BT6	100	0.00 × 10^0^ (0.00 × 10^0^) −	0.00 × 10^0^ (0.00 × 10^0^) −	5.34 × 10^−1^ (3.96 × 10^−2^) +	0.00 × 10^0^ (0.00 × 10^0^) −	**6.24 × 10^−1^ (1.54 × 10^−2^)**
	500	0.00 × 10^0^ (0.00 × 10^0^) −	0.00 × 10^0^ (0.00 × 10^0^) −	5.51 × 10^−1^ (3.63 × 10^−2^) +	0.00 × 10^0^ (0.00 × 10^0^) −	**5.85 × 10^−1^ (1.08 × 10^−2^)**
	1000	0.00 × 10^0^ (0.00 × 10^0^) −	0.00 × 10^0^ (0.00 × 10^0^) −	5.34 × 10^−1^ (3.96 × 10^−2^) +	0.00 × 10^0^ (0.00 × 10^0^) −	**5.43 × 10^−1^ (1.90 × 10^−2^)**
+/−/=		0/45/0	4/40/1	7/37/1	4/41/0	

**Table 3 entropy-25-00561-t003:** IGD Values of the Five Algorithms on the LSMOP Test Suites.

Problem	MOEA/D2	LMEA	IM-MOEA/D	FDV	LSMOEA-TM
LSMOP1	6.94 × 10^0^ (8.21 × 10^−1^) −	1.65 × 10^−1^ (1.82 × 10^−1^) −	2.40 × 10^−1^ (8.69 × 10^−2^) −	2.19 × 10^−1^ (2.59 × 10^−3^) −	**5.62 × 10^−2^ (3.52 × 10^−3^)**
LSMOP2	1.08 × 10^−1^ (4.91 × 10^−3^) −	9.69 × 10^−2^ (8.76 × 10^−2^) −	**6.03 × 10^−2^ (1.08 × 10^−3^) +**	7.28 × 10^−2^ (1.34 × 10^−3^) +	8.63 × 10^−2^ (2.42 × 10^−3^)
LSMOP3	1.69 × 10^1^ (2.32 × 10^0^) −	9.38 × 10^−1^ (1.21 × 10^0^) =	6.96 × 10^−1^ (1.65 × 10^−1^) −	**4.12 × 10^−1^ (6.30 × 10^−2^) +**	6.10 × 10^−1^ (7.95 × 10^−2^)
LSMOP4	3.03 × 10^−1^ (9.56 × 10^−3^) −	1.37 × 10^−1^ (1.02 × 10^−1^) −	9.90 × 10^−2^ (2.49 × 10^−3^) −	1.24 × 10^−1^ (3.35 × 10^−3^) −	**8.87 × 10^−2^ (5.95 × 10^−3^)**
LSMOP5	1.04 × 10^1^ (2.98 × 10^0^) −	3.84 × 10^0^ (2.98 × 10^0^) −	2.29 × 10^−1^ (8.94 × 10^−2^) −	5.41 × 10^−1^ (2.16 × 10^−4^) −	**7.29 × 10^−2^ (3.62 × 10^−3^)**
LSMOP6	1.59 × 10^1^ (6.45 × 10^2^) −	3.08 × 10^1^ (1.23 × 10^2^) −	**1.04 × 10^−2^ (3.21 × 10^−1^) +**	1.18 × 10^−1^ (2.02 × 10^−3^) −	5.26 × 10^−2^ (9.46 × 10^−1^)
LSMOP7	1.58 × 10^0^ (6.60 × 10^−2^) −	1.36 × 10^0^ (1.90 × 10^−1^) −	9.78 × 10^−1^ (7.00 × 10^−2^) −	**9.00 × 10^−1^ (1.22 × 10^−2^) +**	9.17 × 10^−1^ (2.27 × 10^−1^)
LSMOP8	9.26 × 10^−1^ (6.60 × 10^−2^) −	1.08 × 10^−1^ (7.33 × 10^−3^) −	3.51 × 10^−1^ (1.98 × 10^−2^) −	3.60 × 10^−1^ (9.23 × 10^−3^) −	**8.29 × 10^−2^ (4.33 × 10^−3^)**
LSMOP9	4.07 × 10^1^ (9.28 × 10^0^) −	1.29 × 10^0^ (1.16 × 10^0^) −	1.30 × 10^0^ (2.48 × 10^−1^) −	1.19 × 10^0^ (4.06 × 10^−1^) −	**1.82 × 10^−1^ (9.94 × 10^−3^)**
+/−/=	0/9/0	0/8/1	2/7/0	3/6/0	

**Table 4 entropy-25-00561-t004:** CPF Values of the Five Algorithms on the LSMOP Test Suites.

Problem	MOEA/D2	LMEA	IM-MOEA/D	FDV	LSMOEA-TM
LSMOP1	0.00 × 10^0^ (0.00 × 10^0^) −	6.71 × 10^−1^ (2.21 × 10^−1^) −	6.12 × 10^−1^ (7.79 × 10^−4^) −	6.03 × 10^−1^ (1.24 × 10^−1^) −	**8.03 × 10^−1^ (7.08 × 10^−3^)**
LSMOP2	7.41 × 10^−1^ (6.55 × 10^−3^) −	7.70 × 10^−1^ (6.06 × 10^−2^) +	7.95 × 10^−1^ (1.38 × 10^−3^) +	**8.08 × 10^−1^ (1.35 × 10^−3^) +**	7.50 × 10^−1^ (7.16 × 10^−3^)
LSMOP3	0.00 × 10^0^ (0.00 × 10^0^) −	1.46 × 10^−1^ (1.02 × 10^−1^) =	**4.27 × 10^−1^ (1.03 × 10^−1^) +**	1.26 × 10^−1^ (1.16 × 10^−1^) =	1.11 × 10^−1^ (5.89 × 10^−2^)
LSMOP4	4.78 × 10^−1^ (1.06 × 10^−2^) −	7.13 × 10^−1^ (9.47 × 10^−2^) −	7.37 × 10^−1^ (3.49 × 10^−3^) −	**7.62 × 10^−1^ (2.75 × 10^−3^) +**	7.51 × 10^−1^ (1.08 × 10^−2^)
LSMOP5	0.00 × 10^0^ (0.00 × 10^0^) −	1.08 × 10^−1^ (1.63 × 10^−1^) −	3.35 × 10^−1^ (1.66 × 10^−3^) −	4.19 × 10^−1^ (4.33 × 10^−2^) −	**5.05 × 10^−1^ (6.57 × 10^−3^)**
LSMOP6	0.00 × 10^0^ (0.00 × 10^0^) −	1.27 × 10^−1^ (1.57 × 10^−2^) −	**8.54 × 10^−1^ (4.85 × 10^−2^) +**	6.32 × 10^−1^ (4.24 × 10^−3^) −	8.14 × 10^−1^ (2.67 × 10^−2^)
LSMOP7	0.00 × 10^0^ (0.00 × 10^0^) −	1.57 × 10^−1^ (1.92 × 10^−2^) −	6.78 × 10^−1^ (1.71 × 10^−2^) −	**8.49 × 10^−1^ (4.74 × 10^−2^) +**	8.18 × 10^−1^ (1.71 × 10^−2^)
LSMOP8	2.78 × 10^−2^ (1.20 × 10^−3^) −	4.31 × 10^−1^ (1.22 × 10^−2^) −	3.64 × 10^−1^ (1.80 × 10^−3^) −	3.51 × 10^−1^ (9.96 × 10^−3^) −	**4.77 × 10^−1^ (6.62 × 10^−3^)**
LSMOP9	0.00 × 10^0^ (0.00 × 10^0^) −	8.55 × 10^−2^ (6.57 × 10^−2^) −	1.28 × 10^−1^ (4.18 × 10^−2^) −	1.14 × 10^−1^ (2.18 × 10^−2^) −	**2.04 × 10^−1^ (5.25 × 10^−3^)**
+/−/=	0/9/0	1/7/1	3/6/0	3/5/1	

**Table 5 entropy-25-00561-t005:** Average Runtime (s) of Five Algorithms on Some Test Problems.

Problem	D	MOEA/D2	LMEA	IM-MOEA/D	FDV	LSMOEA-TM
DTLZ1	100	168.83	224.63	474.65	243.91	227.60
	300	272.06	384.71	607.00	576.66	352.90
	500	380.38	551.08	615.88	786.35	481.02
DTLZ2	100	157.58	230.39	464.97	240.74	230.78
	300	236.75	370.66	577.62	561.55	346.33
	500	319.52	521.12	562.91	739.27	470.82
UF1	100	165.81	234.19	498.66	251.45	244.10
	300	261.72	407.26	632.01	598.13	379.70
	500	369.13	572.05	625.15	810.61	496.00
UF2	100	175.96	257.42	577.15	271.65	268.19
	300	313.04	460.12	901.47	733.59	458.52
	500	450.95	652.48	808.19	940.94	618.54
WFG1	100	66.41	88.65	216.48	91.92	129.69
	300	270.90	366.31	664.18	529.00	436.69
	500	702.05	918.97	1420.00	1380.80	1192.80
WFG2	100	61.70	85.43	199.48	79.09	101.67
	300	236.76	344.46	564.17	440.87	566.61
	500	586.46	881.83	1113.90	1107.10	1211.00
BT1	100	50.26	70.73	147.84	801.63	119.52
	300	206.88	356.95	559.30	499.84	493.77
	500	568.67	936.16	1245.70	1526.20	1038.00
BT2	100	67.98	91.76	185.13	101.60	151.59
	300	335.92	449.34	619.96	541.92	526.43
	500	886.51	1193.81	1559.92	1732.30	1193.90
LSMOP1	300	196.34	292.76	475.78	418.54	289.54
LSMOP2	300	194.60	307.84	578.82	429.64	296.92
LSMOP3	300	205.80	299.76	571.12	457.34	310.92
LSMOP4	300	205.80	299.76	571.12	457.34	310.92

**Table 6 entropy-25-00561-t006:** Average Runtime (s) of Five Algorithms on Test Suites.

Problem	D	MOEA/D2	LMEA	IM-MOEA/D	FDV	LSMOEA-TM
DTLZ	100	163.47	232.56	468.09	237.06	255.88
	300	255.47	387.12	581.66	553.52	379.97
	500	359.72	554.70	591.68	744.48	499.60
UF	100	170.09	245.30	528.12	262.98	252.13
	300	286.20	440.89	733.59	643.87	437.47
	500	412.96	612.12	693.62	869.50	587.71
WFG	100	67.56	88.97	208.45	84.90	115.65
	300	262.25	359.12	605.24	472.18	475.27
	500	651.99	911.01	1228.20	1203.77	1066.97
BT	100	59.95	82.81	166.00	333.91	117.82
	300	282.39	409.16	600.34	542.06	499.83
	500	812.32	957.18	1061.06	1717.90	1103.87
LSMOP	300	194.35	303.05	570.10	440.48	303.01

## Data Availability

Not applicable.

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
