# Peer review of "A Stable Large-Scale Multiobjective Optimization Algorithm with Two Alternative Optimization Methods"

_entropy, 2023, doi:10.3390/e25040561_

Round 1

Reviewer 1 Report

The papers present a new method for large-scale multi-objective optimization. In the experimental part of the research, the authors verify its performance by comparing it with the performance of other methods. The results are exciting and valuable.

1. Section "Introduction." 

The section is unclear and messy. Discussing the advantages and disadvantages of LSMOEAs with fixed and dynamic grouping strategies is premature because it uses some concepts that have not been explained earlier. Probably, some paragraphs could be moved to the next section. The main novel contributions of the research should be clearly stated at the end. This section should be rewritten, and authors should rethink its content. 

Examples of unexplained or unclear phrases and sentences:

1.1 "The multi-objective optimization problems, whose decision variables are greater than or equal to 100, are called large-scale multi-objective optimization problems (LSMOPs)." What does the expression "variables are greater than or equal to 100" mean? Initially, large-scale optimization generally referred to solving optimization problems involving hundreds or even thousands of decision variables.[7] 

1.2 What do authors mean by "scale of decision space will increase exponentially with the increase of decision variables"? Initially, "with the increase of the number of decision variables, the volume of search space grows exponentially"[7] ("increase of decision variables" is not the same as "increase of the number of decision variables")

1.3 Concepts: "convergence-related variables," "diversity-related decision variables," "initial population," "LMEA," and "CCGDE3" are used without earlier explanation in the discussion about the disadvantages of LSMOEAs with fixed grouping strategies. 

1.4 Why do authors explain "fail to maintain population diversity at the last stage of the evolution process" by saying that "the initial population cannot reflect the distribution of the search space sometimes"? What do they mean when writing about the "search space"? Typically, "search space" means "decision variable space," as in the sentence, "the search space will increase dramatically with the increase of the decision variables." The authors are not sensitive to differences in the use of the term "space" when writing about "decision variable space" and "objective function space." Hence, their comments are unclear.

1.5 Phrase "individual population" is unexplained and misleading. The word "individual" is superfluous and introduces a new unexpected meaning, for example, "individual population" vs. "collective population."

1.6 Fig. 2 is not explained. The sentence "Figure 2 demonstrates that the LMEA fails to maintain population diversity at the last stage of the evolution process" assumes that the reader already understands Fig.2.

2. Section 2.1

Why Eq.(2) is located so far from the first appearance of the phrase "the dominant relation between solutions"? 

3. Section 3.

3.1 The phrase "Two-stage Optimization Method" may be misleading since it suggests that the method consists of a sequence of two stages executed one by one. In fact, the stages are executed alternatively, and the decision on which stage to perform depends on the state of the variable "Diversity_num" and the increase of the population hypervolume.

3.2 The diagram presented in Fig.4 could be improved. I would like you to consider simplification based on introducing two new building blocks: one corresponding to the block labeled as "convergence optimization stage" and the other corresponding to the block labeled as "diversity optimization stage." I would change their labels to "convergence-oriented stage" and "diversity-oriented stage" or maybe "convergence-supporting stage" and "diversity-supporting stage." Then the diagram could be simplified by using these blocks with the respective two labels instead of sequences of blocks with labels "Bayesian-based ...", "Obtain Grouping Results.." and "Update Individual Population..". The lists of steps executed in the new blocks could be explained in the text or the figure below the diagram.

3.3 I would change the order of subsections in Section 3. The presentation and discussion of the diagram in Fig.4 are pleasing. Next, I would explain procedures shared by both stages, like "Bayesian-based Parameter Adjustment" and "Update Individual Population Based.." and explain their connections with evolutionary algorithms (describe main EA components, such as the structure of an individual, rules of population management, perturbation operators). In the last part of Section 3, I would emphasize first that the two new grouping strategies represent novelty and then describe them in detail.

4. Section 4.

I would distinguish two subsections among the first paragraphs at the beginning of Section 4: "The set of benchmark problems" and "Measurement Methodology and plan of experiments." Here, the used PC or servers and software characteristics should be mentioned. Approximate times of execution of experiments with the test cases should also be given.

Reviewer 2 Report

1.       The authors should discuss the distinctions between large-scale multi-objective evolutionary algorithms and popular optimization algorithms like GA and PSO.

2.       The authors should explain in detail what Decision variable grouping strategies mean.

3.       I suggest that the authors provide references to support their claims, particularly in Lines 43-44, 48-50, and other relevant sections. It is essential to back up statements with reliable sources to enhance the credibility and validity of the paper.

Round 2

Reviewer 1 Report

I am satisfied with the improvements concerning my comments. The final version of the paper must be reviewed by a native speaker.
